# Variation in Abundance Ratio of Isoprene and Dipentene Produced from Wear Particles Composed of Natural Rubber by Pyrolysis Depending on the Particle Size and Thermal Aging

**DOI:** 10.3390/polym15040929

**Published:** 2023-02-13

**Authors:** Uiyeong Jung, Sung-Seen Choi

**Affiliations:** Department of Chemistry, Sejong University, 209 Neungdong-ro, Gwangjin-gu, Seoul 05006, Republic of Korea

**Keywords:** tire wear particle (TWP), natural rubber (NR), pyrolysis behavior, wear particle size, thermal aging, crosslink density

## Abstract

Tire wear particles (TWPs) are generated by friction between the road and the tire. TWPs are one of the major microplastics found in environmental samples, such as road dust, particulate matter (PM), and sediment. TWP contents in environmental samples are generally analyzed using the pyrolysis technique. Tire tread compounds of heavy vehicles are usually composed of natural rubber (NR). Isoprene and dipentene are the principal pyrolysis products of NR, and dipentene is employed as the key marker for the determination of the TWP contents. In this study, an NR abrasion specimen was thermally aged, and an abrasion test was performed to obtain the wear particles. The influence of the wear particle size and thermal aging on the pyrolysis behavior of NR was investigated. The isoprene/dipentene ratio exponentially increased as the wear particle size decreased, and it was also increased by the thermal aging of the abrasion specimen. The increased isoprene/dipentene ratio by thermal aging was explained by increasing the crosslink density. Using the relationship between the wear particle size and the isoprene/dipentene ratio, it is possible to estimate the isoprene/dipentene ratio for very small TWP such as PM. The experimental results concluded that the wear particle size and thermal aging affect the formation of the key pyrogenic products, and the influencing factors should be considered for the quantification of TWP contents in the environmental samples.

## 1. Introduction

Friction occurs between the tire tread and road surface to produce tire wear particles (TWPs) while a vehicle is driving. TWPs cause environmental pollution as microplastics. Emissions of TWPs from EU countries and the USA are more than 1.3 and 1.1 million tonnes every year, respectively, while that from the whole world is estimated to be more than 5 million tonnes per year [1,2,3]. The emission of TWPs will increase as the number of vehicles increases. TWPs can enter the atmospheric and aquatic environments. Hence, many studies on the characterization and quantification of TWP in the environment have been performed [4,5,6,7,8,9,10,11].

Tire tread compounds are composed of rubber, filler, and various additives such as crosslinking agents, antioxidants, and processing aids [12,13,14]. Rubbers used for manufacturing tire tread compounds are natural rubber (NR), butadiene rubber (BR), and styrene-butadiene rubber (SBR). Depending on the purpose of the tire, the rubber composition can vary [15,16,17]. The physical and chemical properties of tire tread compounds depend on the formulations [18]. Sulfur vulcanization is used for a crosslinking system of tire tread compounds [19,20,21]. Physical properties such as modulus, hardness, tensile strength, and elongation at the break of a rubber vulcanizate are affected by the crosslink density [22,23].

Friction and heat generated by the wear process age the tire, which affects the properties of the tire tread [24]. Thermally aged tread compounds can generate additional network chains and may increase the crosslink density [25]. These changes are directly related to the lifespan and abrasion behavior of the tire. By thermal aging of a rubber vulcanizate used for tire tread, a real situation for the thermally aged tire can be reflected.

A gas chromatograph equipped with a pyrolyzer (Py-GC) is widely used for the analysis of pyrogenic products formed from polymeric materials, and it is an effective analytical method for tire tread compounds without sample pretreatment [26,27,28,29,30]. Depending on the kind of rubber, characteristic pyrogenic products of monomers and dimers are generated. For example, the most abundant pyrolysis products formed from NR are isoprene and dipentene, corresponding to the monomer and dimer, respectively (Figure 1) [31,32]. Sample size affects pyrolysis behavior, such as pyrolysis rate and kinds and abundances of pyrolysis products [31,33,34,35]. Determination of TWP content in an environmental sample using pyrolysis analysis is performed by the calibration curve [36]. Isoprene and dipentene are used for the analysis of NR components in TWP as the marker. In general, dipentene is used for building a calibration curve for the quantification of NR components in TWP [2,3,36,37]. In addition to the sample size, the crosslink density of a rubber vulcanizate influences the production rates of isoprene and dipentene [38].

There are various sizes of TWPs in road dust [39,40]. In this study, model TWPs were prepared using a model tire tread compound made of NR through an abrasion test, and a single wear particle was pyrolyzed. Variation in abundances of the principal pyrogenic products formed from NR with the wear particle size was investigated. In general, marine transport is used for the intercontinental trade of tires and vehicles equipped with tires. Hence, the thermal aging of tires in the containers naturally proceeds for a long time. If vehicles are parked for quite a while, the tires are also thermally aged during the parking period. The virgin and worn samples were thermally aged to reflect the real situations for a long time at relatively high temperatures. Wear particles produced from the thermally aged samples were also analyzed. The results were supplemented by measuring the crosslink densities of the samples. It is very hard to measure the size of very small TWP as PM exactly. In the present work, the relationship between the pyrolysis behavior of NR and the wear particle size was investigated, and the pyrolysis behavior of very small TWPs was estimated using the relationship.

## 2. Materials and Methods

### 2.1. Materials and Samples

A model tire tread compound was prepared using NR (TSR20, 100 phr), carbon black (N234, 60 phr), processing oil (5 phr), stearic acid (3 phr), zinc oxide (4 phr), anti-degradants (total 4 phr), *N*-tert-butylbenzothiazole-2-sulfenamide (TBBS, 1.1 phr), *N*-(cyclohexylthio)phthalimide (CTP, 0.3 phr), and sulfur (1.6 phr). Mixing was performed in a Banbury-type mixer, and the initial temperatures of the mixer were 110 and 80 °C for the masterbatch (MB) and final mixing (FM) stages, respectively. The abrasion specimens were prepared by curing the rubber compound at 160 °C for the maximum cure time (t_max_) in a compression mold (83 mm diameter and 19 mm thickness). Acetone, tetrahydrofuran (THF), *n*-hexane, and toluene were purchased from Aldrich Co. (Wyoming, IL, USA).

Three samples were prepared (Table 1): (1) untreated sample (sample code: NR0), (2) thermally aged sample (sample code: NRth), and (3) thermally aged sample after pre-abrasion (sample code: NRabth). Thermal aging was performed at 80 °C for 30 days in a convection oven. The aging temperature of 80 °C was determined by considering efficient thermal aging did not cause abnormal effects at high temperatures [41,42,43]. The aging effect at 80 °C might correspond to about 16 times compared to that at 40 °C [44,45].

An abrasion test was performed using a LAT100 tire tread compound tester of the VMI group (Gelderland, The Netherlands). Electro Corundum Disc Grain, 60 of VMI group (Gelderland, The Netherlands), was used as the abrasive disk. The load force was 75 N, and the velocity was 25 km/h. After the abrasion test, the wear particles were collected and separated by size using a sieve shaker of Octagon 200 (Endecotts Co., London, UK). Standard test sieves of 1000, 500, 212, 106, and 63 μm were used. The wear particles were divided into five groups; 63–106, 106–212, 212–500, 500–1000, and larger than 1000 μm.

### 2.2. Morphology and Crosslink Density

Morphologies of the wear particles were observed using an image analyzer (EGVM 35B, EG Tech. Co., Anyang, Republic of Korea). Crosslink densities of the samples were measured by the swelling method [46,47,48]. Three parts of each abrasion sample were cut. Organic additives in the sample were removed by extracting with THF and *n*-hexane for 3 and 2 days, respectively, and the sample was dried for 2 days at room temperature. The weight of the organic materials-extracted sample was measured. The organic materials-extracted sample was soaked in toluene for 2 days at room temperature, and the weights of the swollen samples were measured. The crosslink densities (*X_c_*s) were calculated using the Flory–Rehner Equation (1) [49]
*X_c_* = −[ln(1 − ν_2_) + ν_2_ + χν_2_^2^]/[*V*_1_(ν_2_^1/3^ − ν_2_/2)](1)
where *v*_2_ is the volume fraction of the crosslinked polymer, χ is the interaction parameter between the polymer and solvent, and *V*_1_ is the molar volume of the swelling solvent. The *v*_2_ is obtained by Equation (2)
*v*_2_ = (*m*_2_/ρ_2_)/[(*m*_2_/ρ_2_) + (*m*_1_/ρ_1_)] (2)
where *m*_1_ and *m*_2_ are the solvent and specimen weights at equilibrium swelling, respectively, and ρ_1_ and ρ_2_ are the densities of swelling solvent and unswollen rubber sample, respectively. The interaction parameter of NR with toluene is 0.393 [50].

### 2.3. Pyrolysis-Gas Chromatography (Py-GC)

A furnace-type pyrolyzer of a pyro probe 2000 system with a CDS 1500 interface (Chemical Data System, Oxford, MS, USA) was used. A quartz tube was used for sampling. The sample was pyrolyzed at 520 °C for 10 s under a nitrogen (N_2_) atmosphere. Wear particles with the sizes of 63–106, 106–212, 212–500, 500–1000, and larger than 1000 μm were analyzed. Except for the wear particles of 63–106 μm, each single wear particle was pyrolyzed. For the wear particles of 63–106 μm, three particles were used for one pyrolysis because the size of the one wear particle was too small to pyrolyze efficiently.

The pyrolysis products were separated through gas chromatography (GC) and detected with a flame ionization detector (FID). GC-FID analysis was carried out using a YL 6500 GC system (Younglin Co., Republic of Korea). An HP-5 capillary column (30 m × 0.32 mm, 25 μm film thickness) (Agilent Technology Inc., Santa Clara, CA, USA) was used. Nitrogen (N_2_) was used as the carrier gas, and the flow rate was 3 mL/min. The injector and detector temperatures of GC were 250 °C. The GC oven temperature program was as follows: 30 °C (held for 3 min) to 50 °C at 10 °C/min (held for 3 min), to 180 °C at 10 °C/min (held for 1 min), and to 250 °C at 10 °C/min (held for 3 min).

## 3. Results and Discussion

### 3.1. Wear Particles, Crosslink Density, and Principal Pyrogenic Products

Five wear particles in the one-sieve size range were randomly selected and analyzed. For wear particles in the range of 63–106 μm, three particles were used for one pyrolysis, and five sets were analyzed. Sample sizes of the wear particles were described as two dimensions (multiplication of the long and short axes) and were listed in Table 2. The weight range of the wear particles was 1–1074 μg. Magnified images of the wear particles produced from the NR0 sample are representatively shown in Figure 1. The crosslink density of the abrasion specimen was increased by thermal aging (Figure 2). The crosslink density of a sulfur-cured NR sample is usually increased by thermal aging [51,52]. Measurement errors for the crosslink densities of the aged samples were larger than that of the unaged ones. This can be explained by the heat transfer difference in the outer and inner parts of the sample during thermal aging. The abrasion specimen is relatively thick at 19 mm, and the thermal conductivity of a rubber vulcanizate is not high [53,54]. Hence, heat transfer efficiency might be different depending on the outer and inner parts, which could lead to a difference in the crosslink densities. The crosslink density of the NRth sample was higher than that of the NRabth sample. This may be due to deformation and loss of organic additives by stress during the abrasion test, which might lead to a difference in the initial state of the samples before the thermal aging. Two Py-GC chromatograms of the wear particles of the NR0 sample are representatively shown in Figure 3. Intensities of the isoprene and dipentene for the wear particles larger than 1000 μm were much greater than those for the wear particles of 500–1000 μm because of the difference in the sample sizes.

### 3.2. Variations of the Isoprene/Dipentene Ratios with the Wear Particle Size and Crosslink Density

Variations of the isoprene/dipentene ratios with the wear particle size and crosslink density were examined (Figure 4). The isoprene/dipentene ratio exponentially increased as the wear particle size decreased. The ratio slightly increased until around 1.0 × 10^5^ μm^2^ of the wear particle size and then steeply increased. The increased ratio of isoprene/dipentene with a decrease in the wear particle size can be explained by a difference in the heat transfer efficiencies from the sample cell to the wear particle. By decreasing the sample size, heat will efficiently transfer to the inner part of the sample. This can lead to a higher temperature applied to the small sample rather than the large one. In general, the production rate of a monomer is faster than that of a dimer as the pyrolysis temperature increases [55,56,57,58]. A calibration curve built with the reference samples is used to determine the TWP contents in environmental samples, as introduced previously. Thus, the experimental results can be concluded that the reference samples with particle shape have to be used by considering the difference in heat transfer efficiencies due to the sample sizes in order to reduce the experimental errors.

The isoprene/dipentene ratio tended to increase as the crosslink density increased. Bond strengths of sulfur crosslinks formed in a sulfur-cured rubber sample are much weaker than that of a carbon-carbon single bond (~C–C~) [59]. When thermal energy is applied to the sulfur-crosslinked sample, the sulfur crosslinks must be preferentially dissociated rather than the ~C–C~ bonds in the NR backbone. The sulfur radical in the NR chain formed by the dissociation of the sulfur crosslinks is rearranged to produce isoprene, as shown in Figure 2 [38]. In general, a calibration curve for the quantification of NR contents in environmental samples is built using the reference samples prepared by pure NR. Thus, it should be considered the difference in the isoprene/dipentene ratios between pure NR and sulfur-cured NR.

### 3.3. Relationship between the Isoprene/Dipentene Ratio and the Wear Particle Size

In order to find a good relationship between the isoprene/dipentene ratio and the wear particle size, the isoprene/dipentene ratio was plotted as a function of logarithmic particle size, as shown in Figure 5. Only one datum of the 63–106 μm particles of the NRth sample was excluded because it showed an abnormally large value of the isoprene/dipentene ratio. At the wear particle size of 1.0 × 10^5^ μm^2^ (logarithmic scale = 5.0), the variations were changed at the wear particle size of 1.0 ×10^5^ μm^2^; the increasing rates at the wear particle size smaller than 1.0 × 10^5^ μm^2^ were much greater than those at the particle size larger than 1.0 × 10^5^ μm^2^. For the wear particles smaller than 1.0 × 10^5^ μm^2^, the increasing rates of the isoprene/dipentene ratios were 6.27 × 10^−2^, 6.49 × 10^−2^, and 8.41 × 10^−2^ per 10 μm^2^ of the wear particle size for the NR0, NRth, and NRabth samples, respectively. For the wear particles larger than 1.0 × 10^5^ μm^2^, the increasing rates were 1.14 × 10^−2^, 2.46 × 10^−2^, and 1.88 × 10^−2^ per 10 μm^2^, respectively. The wear particle size of 1.0 × 10^5^ μm^2^ roughly corresponds to the sieve size of 212 μm. For the wear particles larger than 2.0 × 10^6^ μm^2^ (logarithmic scale = 6.3), the isoprene/dipentene ratios were almost the same. Variations of the isoprene/dipentene ratios of the wear particles smaller than 1.0 × 10^5^ μm^2^ with the crosslink density did not show a specific trend, but for the wear particles larger than 1.0 × 10^5^ μm^2^ the increasing rate tended to increase as the crosslink density increased.

### 3.4. Application to PM

In order to quantitatively examine the increasing rate of the isoprene/dipentene ratio with the wear particle size, all data were gathered, as shown in Figure 6. Three cases were curve-fitted; (1) including all data, (2) including data for the wear particles smaller than 1.0 × 10^5^ μm^2^, and (3) including data for the wear particles larger than 1.0 × 10^5^ μm^2^. The three curve-fitted equations are marked in Figure 6. When using the curve-fitted equations for the cases (1) *y* = 0.0141*x*^2^ − 0.187*x* + 0.6328 and (2) *y* = −0.0712*x* + 0.3983 because of clearly an increased trend for the isoprene/dipentene ratio with a decrease in the wear particle size, it is possible to estimate the isoprene/dipentene ratio for smaller particle such as PM. If the size of one PM sample (PM_10_ with a diameter of 10 μm) is substituted to the particle size of 100 μm^2^ (logarithmic scale = 2.0), the isoprene/dipentene ratios will be 0.316 and 0.256 for the curve-fitted Equations (1) and (2), respectively.

As discussed above, the isoprene/dipentene ratio increases as the wear particle size decrease. Some TWPs are in the air as PM, and TWP content in PM is generally quantified using the pyrolysis technique [5,60,61,62,63]. However, it is hard to practically prepare the reference rubber samples for building a calibration curve as small as PM. Hence, quantification results for TWP content in PM may be underestimated if the calibration curve is built using the reference rubber samples larger than PM.

## 4. Conclusions

TWPs were prepared through an abrasion test of the model tire tread compound (NR = 100). The crosslink density of the sample was increased by thermal aging. The wear particles were divided into five groups; 63–106, 106–212, 212–500, 500–1000, and larger than 1000 μm, and the weight range of the wear particles used in this study was 1–1074 μg. Single wear particle was analyzed except for the wear particles of 63–106 μm. The isoprene/dipentene ratio exponentially increased by decreasing the wear particle size, and it was also increased by increasing the crosslink density. When the isoprene/dipentene ratio was plotted as a function of logarithmic wear particle size, the curve-fitted equations were *y* = 0.0141*x*^2^ − 0.187*x* + 0.6328 and *y* = −0.0712*x* + 0.3983 for including all data and including data smaller than 1.0 × 10^5^ μm^2^, respectively. The relationship can be applied to estimate the pyrolysis behavior of smaller particles such as PM. For PM_10_ with a diameter of 10 μm, the isoprene/dipentene ratios of 0.316 and 0.256 can be obtained by applying the curve-fitted equations, respectively. It was found that a reduction in the wear particle size and thermal aging led to an increasing isoprene/dipentene ratio. TWP content in PM can be underestimated when the calibration curve is built using the reference rubber samples larger than PM. It is recommended that the experimental error range for analysis results of TWP content in an environmental sample should be determined by considering the sample states, such as the size, aging history, and crosslink density, especially since the sample size has become smaller.

## Data Availability

The data presented in this study are available on request from the corresponding author.

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
