# Peer review of "Variation in Abundance Ratio of Isoprene and Dipentene Produced from Wear Particles Composed of Natural Rubber by Pyrolysis Depending on the Particle Size and Thermal Aging"

_polymers, 2023, doi:10.3390/polym15040929_

Round 1
Reviewer 1 Report
In the present study, author report, Tire wear particles (TWPs) are generated by friction between the road and tire. TWPs 7 are one of major microplastics found in environmental samples like road dust, particulate matter 8 (PM), and sediment. TWP contents in the environmental samples are generally analyzed using py- 9 rolysis technique.
The work in the manuscript is interesting and satisfactory but some issues need to be revise carefully. The following issue should be considered before publication.
Q-1: The gap in the introduction section of present study and study available in the literature should be highlighted.
Q-2: The author needs to make some changes in the graphical abstract and draw a simple and informative graphical abstract for the study.
Q-3: The Conclusion is very general, the author needs to discuss the main finding in the conclusion section.
Q-4: The abrasion specimen have no any good information, please explain and add more information’s.
Q-5: I suggested the author to follow the guidelines of the journal and arrange the manuscript according to the guidelines.
Q-6: I suggested the author to add the recent references, some references are old. https://doi.org/10.1016/j.surfin.2022.102493, https://doi.org/10.1016/j.eurpolymj.2021.110773
Q-7: The authors must review the English of the manuscript because there are many expressions are ambiguous and unintelligible. Also, a lot of typographical and English error which should be fix.
Author Response
Q1. The gap in the introduction section of present study and study available in the literature should be highlighted.
A1. The sentences “Determination of TWP content in an environmental sample using pyrolysis analysis is performed by the calibration curve [37]. Isoprene and dipentene are used for analysis of NR component in TWP as the marker. In general, dipentene is used for building a calibration curve for quntification of NR component in TWP [2,3,37,38]. Besides the sample size, crosslink density of a rubber vulcanizate influences production rates of isoprene and dipentene [39].” were added in the fourth paragraph along with new references 37-39. In the last paragraph, the sentences “It is very hard to exactly measure the size of very small TWP as PM. In the present work, relationship between the pyrolysis behavior of NR and the wear particle size was investigated, and pyrolysis behavior of very small TWP was estimated using the relationship.” were also added.
Q2. The author needs to make some changes in the graphical abstract and draw a simple and informative graphical abstract for the study.
A2. The graphical abstract was uploaded.
Q3. The Conclusion is very general, the author needs to discuss the main finding in the conclusion section.
A3. It was found that reduction of the wear particle size and thermal aging lead to increasing isoprene/dipentene ratio. The Conclusion part were rewritten. The corrections were marked in blue. The sentences “It is recommended that the experimental error range for analysis results of TWP content in an environmental sample should be determined by considering the sample states such as the size, aging history, and crosslink density. Especially, the sample size gets smaller.” were added.
Q4. The abrasion specimen have no any good information, please explain and add more information’s.
A4. Compound formulation and dimension of the abrasion specimen were already described in the Experimental section 2-1 in detail. The compound formulation was NR (TSR20, 100 phr), carbon black (N234, 60 phr), processing oil (5 phr), stearic acid (3 phr), zinc oxide (4 phr), antidegradants (total 4 phr), N-tert-butylbenzothiazole-2-sulfenamide (TBBS, 1.1 phr), N-(cyclohexylthio)phthalimide (CTP, 0.3 phr), and sulfur (1.6 phr). The dimension was 83 mm diameter and 19 mm thickness.
Q5. I suggested the author to add the recent references, some references are old. https://doi.org/10.1016/j.surfin.2022.102493, https://doi.org/10.1016/j.eurpolymj.2021.110773
A5. The two recommended references are not related to this study. They are about conducting polymers. The first reference (https://doi.org/10.1016/j.surfin.2022.102493) is study on fabrication of MXene (Ti2C3Tx) based conducting polymer materials, and the second reference (https://doi.org/10.1016/j.eurpolymj.2021.110773) is study on conducting polymer-based nanocomposites. Recent references of 37, 38, 54, and 55 were newly cited.
Q6. The authors must review the English of the manuscript because there are many expressions are ambiguous and unintelligible. Also, a lot of typographical and English error which should be fix.
A6. English of the manuscript was revised, and the corrections were marked in blue.
Reviewer 2 Report
In this work, authors reported on the pyrolysis of pre-aged NR. Some comments should be addressed by the authors before acceptance:
- What is the aim and the scientific significance of the study? This should be clarified in the introduction section.
- Avoid abbreviations in the title, i.e. NR.
- Avoid the use of “We” in the manuscript (see line 64).
- Isoprene/dipentene ratio from pyrolysis of NR was analyzed but the purpose, practical application or scientific impact of this determination is not clear. This should be clarified in the introduction section.
- As authors state in the introduction section (line 53), it is well known that sample size affects pyrolysis behavior of NR (references 33-36), consequently the results shown are foreseeable. Consequently, I cannot find the novelty of this work. The original value of the study and how it contributes to the literature, what differentiate their work from the previously published ones, should be explained clearly.
- Authors should justify that the thermally aged conditions selected (80 ºC for 30 days in a convection oven) reflect the aging of rubber in real situation (in which real situation are they thinking?). In this regards, authors should prove (by references or standards) that the proposed method is adequate to properly simulate this real situation.
- According to Figure 2, crosslink density of the NTth sample was higher than that of the NRabth sample. Authors should justify this behavior. Also, maybe, statistical analysis is needed to signify the results of Figure 2.
- Five sets were analyzed by Py-GC, so an average and standard deviation should be expected for isoprene/dipentene ratio. Are the results plotted in Figures 4, 5 and 6 showing the average value of isoprene/dipentene ratio? Also, standard deviation should be indicated in Figures 4, 5 and 6.
- In line 157, authors mention that “a calibration curve is used to determine TWP contents in environmental samples”. It is not clear to me if you can determine either TWP contents or TWP particle sizes. Authors should clarify.
- In Figure 5, I cannot see differences between the three samples, it seems that only the particle size has influence in the isoprene/dipentene ratio regardless the sample tested. Should not be also influence of the type of sample? Authors should comment on this.
- It is not clear how the authors want to extrapolate this work to real samples (i.e. road dust or sediments). How to deal with the interference of other substances? This idea should be more clear in the discussion of the paper.
- The conclusion section collects statements of the discussion section and do not address future perspectives for scientific research and possible practical applications. Please, rewrite it.
Author Response
Q1. What is the aim and the scientific significance of the study? This should be clarified in the introduction section.
A1. In this study, model TWPs were prepared using a model tire tread compound made of NR through abrasion test and a single wear particle was pyrolyzed. Variation in abundances of the principal pyrogenic products formed from NR with the wear particle size was investigated. Relationship between the pyrolysis behavior of NR and the wear particle size was examined using the experimental results, and pyrolysis behavior of very small TWP was estimated by the relationship. In the last paragraph, the sentences “It is very hard to exactly measure the size of very small TWP as PM. In the present work, relationship between the pyrolysis behavior of NR and the wear particle size was investigated, and pyrolysis behavior of very small TWP was estimated using the relationship.” were added.
Q2. Avoid abbreviations in the title, i.e. NR.
A2. “NR wear particles composed of natural rubber” was changed to “NR wear particles composed of natural rubber”
Q3. Avoid the use of “We” in the manuscript (see line 64).
A3. The sentence “We thermally aged the virgin and worn samples to reflect the real situations.” was changed to “The virgin and worn samples were thermally aged to reflect the real situations for a long time at relatively high temperature.”.
Q4. Isoprene/dipentene ratio from pyrolysis of NR was analyzed but the purpose, practical application or scientific impact of this determination is not clear. This should be clarified in the introduction section.
A4. The sentences “Isoprene and dipentene are used for analysis of NR component in TWP as the marker. In general, dipentene is used for building a calibration curve for quntification of NR component in TWP [2,3,37,38].” were added in the fourth paragraph along with new references 37 and 38. In the last paragraph, the sentences “It is very hard to exactly measure the size of very small TWP as PM. In the present work, relationship between the pyrolysis behavior of NR and the wear particle size was investigated, and pyrolysis behavior of very small TWP was estimated using the relationship.” were added.
Q5. As authors state in the introduction section (line 53), it is well known that sample size affects pyrolysis behavior of NR (references 33-36), consequently the results shown are foreseeable. Consequently, I cannot find the novelty of this work. The original value of the study and how it contributes to the literature, what differentiate their work from the previously published ones, should be explained clearly.
A5. Compared to other reports, we analyzed wide range of small-sized samples of 0.7´104 mm2 – 243.6´104 mm2 (1 – 1074 mg) with form of tire wear particles. This content was described and discussed in the Experimental and Results and Discussion parts. Relationship between the pyrolysis behavior of NR and the wear particle size was found, and the relationship was applied to the smaller TWP like PM. The sample size was listed in Table 2. The sentence “Weight range of the wear particles was 1 – 1074 mg.” was added in the section 3-1.
Q6. Authors should justify that the thermally aged conditions selected (80 ºC for 30 days in a convection oven) reflect the aging of rubber in real situation (in which real situation are they thinking?). In this regards, authors should prove (by references or standards) that the proposed method is adequate to properly simulate this real situation.
A6. In the Introduction part, Lines 63-64 were changed to “The virgin and worn samples were thermally aged to reflect the real situations for a long time at relatively high temperature.”. In the section 2-1, the sentences “The aging temperature of 80ºC was determined by considering efficiently thermal aging not to happen abnormal effect at high temperature [42-44]. The aging effect at 80ºC might correspond to about 16 times compared to that at 40ºC [45,46].” were added.
Q7. According to Figure 2, crosslink density of the NTth sample was higher than that of the NRabth sample. Authors should justify this behavior. Also, maybe, statistical analysis is needed to signify the results of Figure 2.
A7. Measurement of the crosslink densities was performed by three parts of each abrasion sample. The sentence “Three parts of each abrasion sample were cut.” was added in the section 2-2. In the Results and Discussion part, the sentence “This may be due to deformation and loss of organic additives by stress during the abrasion test, which might lead to difference in the initial states of the samples before the thermal aging.” was added in the section 3-1. The sentences “This may be due to deformation and loss of organic additives by stress during the abrasion test, which might lead to difference in the initial states of the samples before the thermal aging. Measurement errors for the crosslink densities of the aged samples were larger than that of the unaged one. This can be explained by heat transfer difference in the outer and inner parts of the sample during the thermal aging. The abrasion specimen is relatively thick of 19 mm and thermal conductivity of a rubber vulcanizate is not good [54,55]. Hence, heat transfer efficiency might be different depending on the outer and inner parts, which could lead to difference in the crosslink densities.” were also added in the section 3-1.
Q8. Five sets were analyzed by Py-GC, so an average and standard deviation should be expected for isoprene/dipentene ratio. Are the results plotted in Figures 4, 5 and 6 showing the average value of isoprene/dipentene ratio? Also, standard deviation should be indicated in Figures 4, 5 and 6.
A8. For wear particles in the range of 63 – 106 mm, three particles were used for one pyrolysis and five sets were analyzed. The standard deviation was too small compared to the scale of x-axis so it was covered by the symbol.
Q9. In line 157, authors mention that “a calibration curve is used to determine TWP contents in environmental samples”. It is not clear to me if you can determine either TWP contents or TWP particle sizes. Authors should clarify.
A9. In the Introduction part, the sentences “Determination of TWP content in an environmental sample using pyrolysis analysis is performed by the calibration curve [37]. Isoprene and dipentene are used for analysis of NR component in TWP as the marker. In general, dipentene is used for building a calibration curve for quntification of NR component in TWP [2,3,37,38].” were added to clarify about the calibration curve along with the new references 37 and 38. The sentence “a calibration curve is used to determine TWP contents in environmental samples” was changed to “A calibration curve built with the reference samples is used to determine the TWP contents in environmental samples as introduced previously”.
Q10. In Figure 5, I cannot see differences between the three samples, it seems that only the particle size has influence in the isoprene/dipentene ratio regardless the sample tested. Should not be also influence of the type of sample? Authors should comment on this.
A10. Clear differences between the three samples were not observed. However, for the wear particles larger than 1.0´105 mm2 the increasing rate tended to increase as the crosslink density increased. This was discussed in the section 3-3.
Q11. It is not clear how the authors want to extrapolate this work to real samples (i.e. road dust or sediments). How to deal with the interference of other substances? This idea should be more clear in the discussion of the paper.
A11. The isoprene/dipentene ratio clearly increased as the particle size decreased, and the proper curve-fitted equations could be obtained. For this reason, it was possible to extrapolate to the smaller particle. The sentence in the section 3-4 “When using the curve-fitted equations for the cases (1) y = 0.0141x2 - 0.187x + 0.6328 and (2) y = -0.0712x + 0.3983, it is possible to estimate the isoprene/dipentene ratio for smaller particle such as PM.” was changed to “When using the curve-fitted equations for the cases (1) y = 0.0141x2 - 0.187x + 0.6328 and (2) y = -0.0712x + 0.3983 because of clearly increased trend for the isoprene/dipentene ratio with decrease in the wear particle size, it is possible to estimate the isoprene/dipentene ratio for smaller particle such as PM.”.
Q12. The conclusion section collects statements of the discussion section and do not address future perspectives for scientific research and possible practical applications. Please, rewrite it.
A12. It was found that reduction of the wear particle size and thermal aging lead to increasing isoprene/dipentene ratio. The Conclusion part were rewritten. The corrections were marked in blue. The sentences “It is recommended that the experimental error range for analysis results of TWP content in an environmental sample should be determined by considering the sample states such as the size, aging history, and crosslink density. Especially, the sample size gets smaller.” were added.
Round 2
Reviewer 1 Report
I suggested to accept in the present form
Reviewer 2 Report
The authors have addressed properly the comments.
Thank you.